# Complete Revascularization of Multivessel Coronary Artery Disease Does Not Improve Clinical Outcome in ST-Segment Elevation Myocardial Infarction Patients with Reduced Left Ventricular Ejection Fraction

**DOI:** 10.3390/jcm9010232

**Published:** 2020-01-15

**Authors:** Jeehoon Kang, Chengbin Zheng, Kyung Woo Park, Jiesuck Park, Taemin Rhee, Hak Seung Lee, Jung-Kyu Han, Han-Mo Yang, Hyun-Jae Kang, Bon-Kwon Koo, Hyo-Soo Kim

**Affiliations:** Department of Internal Medicine and Cardiovascular Centre, Seoul National University Hospital, Seoul 03080, Korea; medikang@gmail.com (J.K.); sengbin331@snu.ac.kr (C.Z.); cardio.jspark@gmail.com (J.P.); imcrtm@gmail.com (T.R.); cardiolee@gmail.com (H.S.L.); hpcrates@gmail.com (J.-K.H.); hanname@gmail.com (H.-M.Y.); nowkang@snu.ac.kr (H.-J.K.); bkkoo@snu.ac.kr (B.-K.K.); hyosoo@snu.ac.kr (H.-S.K.)

**Keywords:** percutaneous coronary intervention, ST-segment elevation myocardial infarction, complete revascularization, infarct-related artery only treatment, multivessel disease, left ventricular ejection fraction

## Abstract

The benefit of complete revascularization (CR) in ST-segment elevation myocardial infarction (STEMI) patients with left ventricular (LV) dysfunction is uncertain. A total of 1314 STEMI patients with multivessel coronary artery disease were analyzed. CR was defined angiographically and by a residual Synergy between PCI with Taxus and Cardiac Surgery trial (SYNTAX) score (SS) <8. Patients with a left ventricular ejection fraction (LVEF) <40% were classified as the reduced LVEF group. The major study endpoints were patient-oriented composite outcome (POCO) and cardiac death during three-year follow-up. Overall, patients that received angiographic CR (579 patients, 44.1%) had significantly lower three-year clinical events compared with incomplete revascularization (iCR). CR reduced three-year POCO and cardiac death rates in the preserved LVEF group (POCO: 13.2% vs. 21.9%, *p* < 0.001, cardiac death: 1.8% vs. 6.5%, *p* < 0.001, respectively) but not in the reduced LVEF group (POCO: 26.0% vs. 33.1%, *p* = 0.275, cardiac death: 15.1% vs. 19.0%, *p* = 0.498, respectively). Multivariate analysis showed that CR significantly reduced three-year POCO (hazard ration (HR) 0.59, 95% confidence interval (CI) 0.43–0.82) and cardiac death (HR 0.34, 95% CI 0.14–0.80), only in the preserved LVEF group. Additionally, the results were corroborated using the SS-based CR definition. In STEMI patients with multivessel disease, CR did not improve clinical outcomes in those with reduced LVEF.

## 1. Introduction

Current guidelines advocate percutaneous coronary intervention (PCI) for non-culprit arteries in ST-segment elevation myocardial infarction (STEMI) patients who are hemodynamically stable [1,2,3]. The recommendation is supported by four recent randomized clinical trials (RCT) that confirmed the beneficial effect of complete revascularization (CR) in multivessel STEMI patients [4,5,6,7]. In these studies, CR which was achieved through either a one-step or staged procedure, improved clinical outcomes compared with incomplete revascularization (iCR) by 45–65% at follow-up ranging from one to three years, although the primary endpoints were slightly different according to each study.

Previous RCTs had strict inclusion criteria, only including patients who were hemodynamically stable, and patients with few clinical risk factors. In real world practice, however, we often encounter patients with various clinical problems and risk factors, and there is uncertainty as to whether we can extrapolate the beneficial effects of CR in these patients. In particular, the Culprit Lesion Only PCI versus Multivessel PCI in Cardiogenic Shock (CULPRIT-SHOCK) trial showed that CR could not reduce mortality in STEMI patients with cardiogenic shock [8], stressing that additional evidence is needed for CR in high-risk STEMI patients.

Reduced left ventricular ejection fraction (LVEF) is associated with increased mortality in STEMI patients [9,10], and CR may not have profound beneficial effects in this subgroup since these patients are at increased risk for sudden cardiac death, ventricular arrhythmia, and death from progression of heart failure. In fact, a previous registry-based study showed that CR compared with iCR did not improve clinical outcome in coronary artery disease patients with LV ejection fraction <40% [11]. To the best of our knowledge, there are no studies in STEMI patients that address whether CR improves clinical outcome in patients with reduced LVEF. Therefore, in our study, we evaluated the clinical outcomes between CR and iCR in STEMI patients with multivessel disease, according to baseline LVEF.

## 2. Methods

An expanded description of the study method is presented in the Appendix A.

### 2.1. Study Population

Our study was based on the ‘Grand DES registry’ (drug-eluting stent) which is a pooled database of five nation-wide prospective registries from Korea of patients receiving DESs for coronary artery diseases, from 1 January 2004 to 31 November 2014 in 55 centers in Korea. Individual patient data were pooled from the Efficacy of Xience/Promus versus Cypher in rEducing Late Loss after stENTing (EXCELLENT) registry, Efficacy and Safety of Xience in Coronary arEry Disease aLL-comers After stENTing Using the PRIME Platform (EXCELLENT-PRIME) registry, Harmonizing Optimal Strategy for Treatment of coronary artery disease using a RESOLute INTEgrity (HOST-RESOLINTE) registry, Registry to Evaluate the Efficacy of Zotarolimus-Eluting Stent (RESOLUTE-Korea) registry, and the Harmonizing Optimal Strategy for Treatment of coronary artery disease using a BIOLIMUS A9-eluting stent (HOST-BIOLIMUS) registry, including 17,286 patients. After index PCI, clinical follow-ups were performed up to three years. An expanded description of the included registries is presented in the Appendix A. The study complied with the provisions of the Declaration of Helsinki, and the study was approved by the institutional review board at each center. All patients provided written informed consent (ID: NCT03507205).

### 2.2. Completeness of Revascularization and Calculation of the SYNTAX Score

For evaluation of CR and calculation of the SYNTAX score (SS), quantitative analysis of baseline coronary angiographic images was performed by three specialized quantitative coronary angiography (QCA) technicians at the Seoul National University Hospital Cardiovascular Clinical Research Center Angiographic Core Laboratory, who were blinded to all clinical data, presentation, implanted stents, and outcomes. In the event of disagreement, the lesion was reviewed and a final decision was established by consensus. The core lab was validated with SS calculation, showing measurement correlation above 95% [12].

To evaluate the angiographic completeness of revascularization, angiographic images were retrospectively evaluated. Angiographic CR was defined as the treatment of any lesion with more than 70% diameter stenosis in vessels ≥2.5 mm as estimated on the diagnostic angiogram, leaving no residual significant angiographic stenosis. Also, due to the various definitions of CR, and due to the fact that it is not always feasible to completely revascularize multivessel diseases, we calculated the SYNTAX score (SS) system to quantify the degree of revascularization. The SS was calculated by visually assessing all coronary lesions with a diameter stenosis ≥50% in vessels >1.5 mm in diameter, using the SS algorithm, which is available on the SYNTAX score website. Baseline SS was defined as the SS at initial coronary angiography, while the residual SS (rSS) was calculated as the SS after index PCI. SS-based CR was defined as a rSS of less than 8, which was the definition used in previous studies [13].

### 2.3. Echocardiographic Study

For each patient, echocardiography was performed during admission, mostly after the acute phase when the patient was clinically stable. For the LV systolic and diastolic function, dimensions were assessed according to international guidelines [14]. Reduced LVEF was defined as LV ejection fraction less than 40%, while those with LV ejection fraction ≥40% were classified as the preserved LVEF group.

### 2.4. End Points

Clinical follow-up was performed up to three years (median: 1123 days, interquartile range 1078–1137 days). The primary analysis outcome was patient-oriented composite outcome (POCO; a composite of all cause death, any myocardial infarction, and any revascularization) and the key secondary analysis endpoint was cardiac death, both at three years.

### 2.5. Statistical Analyses

Data are presented as numbers and frequencies for categorical variables and as mean ± SD for continuous variables. Clinical and procedural characteristics were compared between patients experiencing clinical events, defined as endpoints. For comparison among groups, χ^2^ or the Fisher exact test (when any expected count was <5 for a 2 × 2 table) for categorical variables and unpaired Student *t* test or one-way analysis of variance for continuous variables were applied. To estimate the independent factors on endpoints, a multivariable Cox proportional hazards regression model using a backward elimination algorithm and 0.05 as the significance level was performed. In addition, the Cox proportional hazard regression in a propensity-score matched cohort (standardized mean difference of variables was <10%) and inverse probability weighted (IPW) Cox proportional hazard regression were also performed. Event rates were calculated based on Kaplan–Meier censoring estimates, and the log-rank test was used to compare between CR and iCR groups. All probability values were two-sided and *p*-values <0.05 were considered statistically significant. Statistical tests were performed using SPSS, V23 (SPSS Inc., Chicago, IL, USA) and R programming language, version 3.4.1 (R Foundation for Statistical Computing, Vienna, Austria).

## 3. Results

### 3.1. The Beneficial Effect of CR in STEMI Patients

From the Grand-DES registry, 1314 STEMI patients with multivessel disease were analyzed in the current study. Among the total population, CR was achieved in 579 patients (44.1%), while iCR was achieved in 735 patients (55.9%) (Figure 1). Among the CR population, 97 patients (16.8%) received a staged procedure to achieve CR. Baseline characteristics of the CR and iCR group are shown in Table 1. The iCR group had more risk factors, such as old age, hypertension, and chronic renal failure, and showed a higher coronary complexity. During revascularization, more drug eluting stents with a longer total length and a smaller minimal stent diameter were used in the CR group. The baseline SYNTAX score was higher in the iCR group (15.6 ± 8.0 vs. 20.3 ± 8.9, *p* < 0.001, CR vs. iCR), while the delta SYNTAX score was higher in the CR group (13.8 ± 7.8 vs. 11.5 ± 7.2, *p* < 0.001, CR vs. iCR). The discharge medication pattern was similar between the CR and iCR groups.

Regarding the three-year clinical outcomes, CR was associated with a significantly lower rate of three-year POCO compared to iCR (14.9% (86/579) vs. 24.4% (179/735), *p* < 0.001, CR vs. iCR), which was mainly driven by a decrease in cardiac death (2.9% (17/579) vs. 9.3% (68/735), *p* < 0.001, CR vs. iCR). Multivariable Cox regression analysis, propensity score matched analysis, and inverse probability weighting adjusted analysis all consistently showed CR significantly reduced the risk of POCO and cardiac death compared with iCR (Appendix A). The survival curve of POCO and cardiac death according to CR is shown in Figure 2.

### 3.2. CR vs. iCR According to Baseline LVEF

To evaluate whether the beneficial effect of CR was preserved regardless of the baseline LVEF, we analyzed the effect of CR according to the presence of baseline LV dysfunction. There were 236 patients (18%) in the reduced LVEF group and 1078 patients (82%) in the preserved LVEF group. The baseline demographics, angiographic findings, and laboratory findings according to LVEF are summarized in Appendix A. Patients with reduced LVEF compared with those without tended to be sicker with a greater number of risk factors (i.e., older in age, higher proportion of congestive heart failure, chronic renal failure, and anemia). Angiographic characteristics showed that the reduced LVEF group had more patients with three-vessel disease, had more calcified lesions, used smaller diameter stents, and had a higher baseline SS and rSS. While CR was more frequently achieved in the preserved LVEF group (rate of CR: 506/1078 (46.9%) vs. 73/236 (30.9%), *p* < 0.001, for the preserved LVEF vs. reduced LVEF group).

At three-year clinical follow-up, POCO rates were significantly lower in the CR group compared with the iCR group in the preserved LVEF group (POCO rate: 67/506 (13.2%) vs. 125/572 (21.9%), *p* < 0.001, in the CR and iCR group, respectively). However, the difference was attenuated in the reduced LVEF group resulting in no significant differences (POCO rate: 19/73 (26.0%) vs. 54/163 (33.1%), *p* = 0.275, for CR vs. iCR group). This trend was consistent for three-year cardiac death with significantly lower cardiac death rates with CR in only the preserved LVEF group. Upon multivariate analysis, iCR was an independent predictor of three-year POCO and cardiac death in only the preserved LVEF group (Table 2, Figure 3). In particular, there was a significant interaction for cardiac death between the effect of CR and the presence of LV dysfunction (P for interaction = 0.036, Appendix A). As seen in the forest plot (Appendix A), the effect of CR was not significantly different along various subgroups. Landmark analysis from 30 days also showed that CR reduced POCO both in the early phase (<30 days) and the late phase (≥30 days to three years) only in the preserved LVEF group (Appendix A).

### 3.3. Corroboration Using SS-Based CR

Due to the various definitions of CR, we calculated the rSS and analyzed the clinical impact of SS-based CR (rSS <8) vs. SS-based iCR (rSS ≥8). The three-year POCO and cardiac death were significantly more common in those with a SS-based iCR (POCO: 17.3% vs. 27.8%, *p* < 0.001; cardiac death: 4.4% vs. 12.8%, *p* < 0.001, in the SS-based CR vs. iCR, respectively). On multivariate analysis, SS-based iCR was again an independent risk factor for three-year clinical events (Table 3).

When stratified into subgroups according to LVEF, SS-based CR and iCR showed distinct effects according to the presence of LV dysfunction. SS-based CR was associated with a lower three-year POCO and cardiac death, only in those with preserved LVEF, while there were no significant differences in the three-year clinical outcomes between SS-based CR and iCR in the reduced LVEF group (Figure 4).

## 4. Discussion

We analyzed the effect of CR on the three-year outcome of STEMI patients with multivessel disease. Patients were divided according to the presence of baseline LV dysfunction. Our main findings are as follows: (1) Overall, in STEMI patients with multivessel coronary artery disease, CR was associated with lower rates of POCO and cardiac death at three years. (2) When divided according to the presence of LV dysfunction, the beneficial effect of CR was only present in those with preserved LVEF. In patients with reduced LVEF, there were no significant differences between CR and iCR for both POCO and cardiac death. Multivariable analysis confirmed that CR was an independent protective factor of POCO only in those with preserved LVEF, which was driven by a decrease in cardiac death. (3) Our findings were concordant using both an angiographic definition of CR and a SS-based definition of CR.

### 4.1. CR in STEMI Patients

Up to now, four RCTs have compared CR vs. iCR in STEMI patients with multivessel coronary disease. Two RCTs used angiography-guided revascularization, and reported lower rates of major adverse cardiac events in the CR group which performed routine revascularization of the non–infarct-related coronary arteries [4,6]. Two other studies performed fractional flow reserve (FFR) guided treatment of non–infarct-related coronary arteries and also reported a significant reduction in major adverse cardiac events, supporting the beneficial effect of CR [5,7]. Taken together, these studies consistently showed that additional PCI to non-culprit arteries could decrease adverse clinical outcomes by up to 45–65%. In the present study, concurrent with previous observations, CR improved both POCO and cardiac death in the overall STEMI multivessel coronary disease population. However, the major caveat of the previous RCTs was the fact that it excluded high-risk patients, such as those with cardiogenic shock, patients with renal impairment, patients who had undergone previous coronary artery bypass graft surgery (CABG), and those with a life expectancy less than the duration of the trial. This limits the generalizability of the results of these studies. Since much of the dilemma in clinical decision-making occurs when we have to make a decision on whether to perform CR in these patients, we need more studies that address the effects of CR on these patients. One of the key studies that addressed such a population was the CULPRIT-SHOCK trial, where the effect of CR was studied in STEMI patients with cardiogenic shock. The trial showed that CR failed to reduce one-year mortality in this high-risk population [8], suggesting that we need to study whether CR actually improves the outcome in other high-risk populations.

Regarding baseline LVEF, previous trials have reported that up to 20% of STEMI patients have LV dysfunction [15], while these patients are at higher risk for a future event after treatment of STEMI. This is due to increases in risk of fatal arrhythmic events, thrombotic events, and aggravation of heart failure [16,17]. In fact, cardioverter-defibrillator implantation has been shown to reduce mortality in STEMI patients with reduced LVEF, and therefore is recommended by the guidelines for primary prevention of sudden cardiac death [18]. Moreover, a previous study showed that PCI is superior to CABG in patients with lower LVEF [19]. This may be related to higher surgical risk in patients with lower LVEF and loss of major advantages of CABG. Regarding this analysis, we can stress the clinical importance of PCI in STEMI patients with lower LV dysfunction. However, those with significant LV dysfunction were excluded in the previous RCTs. Moreover, a previous study based on New York’s PCI reporting system analyzed that there were no significant differences in 18-month mortality between CR and iCR in patients with LVEF <40%. [11] Therefore, we wanted to evaluate whether the beneficial effect of CR in multivessel STEMI patients would be sustained in those with reduced LVEF.

### 4.2. CR in STEMI Patients with Reduced LVEF (Moderate to Severe LV Dysfunction)

Regarding CR, many mechanisms can explain the beneficial effect of additional non-culprit intervention. This includes improvement in myocardial salvage by revascularization of hibernating myocardium and increase of blood flow to watershed areas. However, our study showed that in those with reduced baseline LVEF, the theoretical benefits of CR did not lead to improvement of the clinical outcome. Interestingly, a previous meta-analysis suggested that revascularization was superior to medical treatment only in the presence of a viable myocardium in patients with coronary artery disease and significant LV dysfunction [20]. Our results are in line with this study, because CR was performed without any evaluation of myocardial viability. Although some studies have evaluated the association of FFR and myocardial viability, FFR values alone have limitations in assessing viability, and other non-invasive methods such as cardiac MRI and PET scanning are necessary to accurately assess myocardial viability.

In the present study, we found that CR reduced three-year POCO by 42%, which is a similar result to that of previous RCTs. However, this was only observed in those with preserved LVEF and not in those with reduced LVEF. The reduction in clinical events were mainly driven by a significant reduction in cardiac death, while any MI and any revascularization events were also numerically lower in the CR group. When comparing the effect of CR in those with and without significant LV dysfunction, not only was the statistical difference between CR and iCR not significant in those with reduced LVEF, but the numerical spot HR of CR (vs. iCR) was less protective in the group with reduced LVEF. This trend was found in not only POCO but other components such as all-cause death, cardiac death, and target lesion failure (TLF), suggesting that the effect of CR was not as prominent in these patients. This may be explained by the increased risk of arrhythmia in STEMI patients with significant LV dysfunction, leading to more cardiac death events [17,18]. Also, the increased thrombotic risk in LV dysfunction patients lead to a higher risk of stent-related ischemic events such as myocardial infarction and stent thrombosis [16,21]. By landmark analysis, we showed that the lack of benefit of CR in those with reduced LVEF was consistent both within the initial 30 days after PCI, and from the 30 days to three years. A recent trial showed that CR was superior to culprit-only PCI in STEMI patients with LVEF <45% [22]. Although the results seem to be inconsistent with our analysis, a few points should be noted before comparison. Compared to this study, our study population included patients presenting as cardiogenic shock, included patients with more clinical risk factors (i.e., hypertension, diabetes, chronic renal failure, etc.), and had a higher baseline SYNTAX score. Also, the randomization point was after the index PCI within 72 h, implying that very high-risk patients were not enrolled in the study. Collectively, our study was based on real-world patients, including patients with a higher risk profile. The markedly different study population, along with the intrinsic difference between RCTs and registry-based studies, could have contributed to the different results from our study.

### 4.3. Confirmation of the Effect of CR Using a SYNTAX Score-Based Definition of CR

To confirm our findings using a more quantifiable definition of CR, we analyzed our data using a SS-based definition of CR [12]. We calculated the rSS post-PCI by quantifying the residual disease summing the points of each coronary lesion with ≥50% stenosis in vessels ≥1.5 mm in diameter [23]. We used previous studies’ definition of reasonable incomplete revascularization of a rSS <8 [24], to evaluate the effect of SS-based CR in our population. As was the results with angiographic CR, SS-based CR was only beneficial in STEMI patients with preserved LVEF.

### 4.4. Limitations

Our study has several limitations. First, this study was an observational analysis of a prospective registry; therefore, treatment (CR vs. iCR) was not randomized and decided by the treating physician. Second, adjunctive therapy post-PCI was also determined by the treating physician. These limitations leave open the possibility of selection bias and treatment bias. Although we used multiple statistical models to correct for possible biases and confounders, the possibility of unforeseen confounders affecting the outcome cannot be completely ruled out. Accordingly, our study results may not be the sole consequence of CR or iCR. Third, CR was evaluated by angiography, without functional studies such as FFR. Fourth, although current guidelines recommend potent new generation P2Y12 inhibitors such as ticagrelor and prasugrel in STEMI patients, a majority of our study population was prescribed clopidogrel. This was due to the delayed approval of these agents in Korea, which was in 2013. Finally, echocardiography was performed when the patient was clinically stabilized, before discharge. However, LVEF assessment immediately after PCI may not reliably predict chronic LV dysfunction, especially in STEMI patients [25]. Moreover, a recent study suggested that MI patients who have an improved LVEF during the follow-up may have better clinical outcomes than those with no improvement [26]. The absence of a follow-up LVEF assessment could act as a bias for our study.

## 5. Conclusions

Among patients with STEMI and multivessel coronary artery disease, the benefit of CR overall, was confirmed in a large-scale prospective registry. However, when divided into those with and without significant LV dysfunction, the beneficial effect of CR was not observed in those with reduced LVEF. Our results suggest that CR of multivessel disease in STEMI patients should be attempted more actively in those with preserved LVEF.

## Figures and Tables

**Figure 1 jcm-09-00232-f001:**
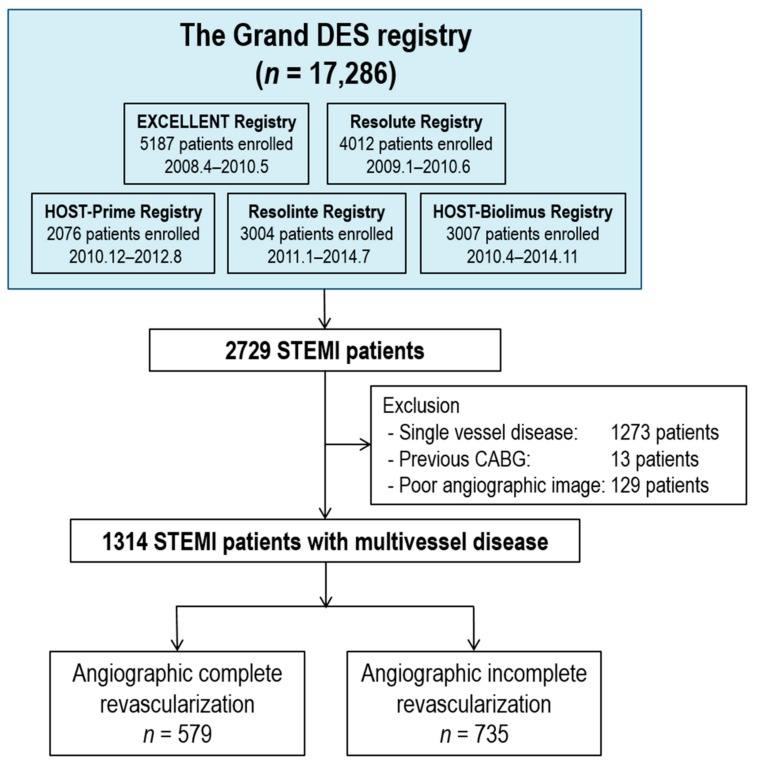
Study population. Abbreviations: CABG, coronary artery bypass graft surgery; STEMI, ST-segment elevation myocardial infarction.

**Figure 2 jcm-09-00232-f002:**
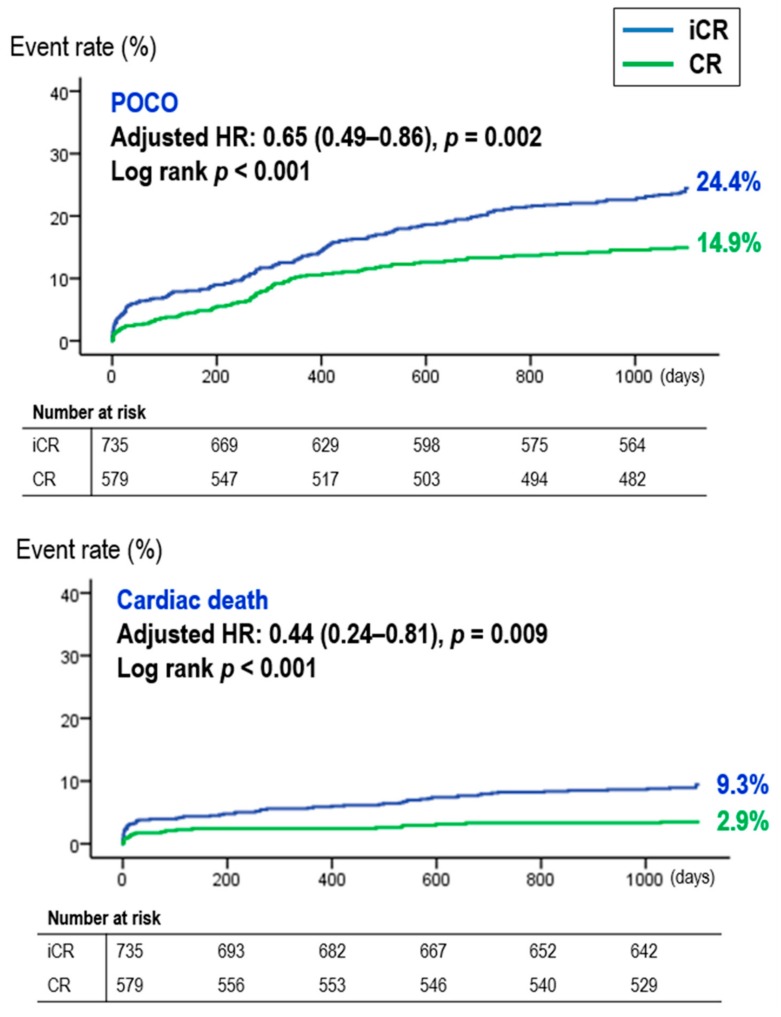
Survival curves during the three-year follow up period. Abbreviations: CR, complete revascularization; HR, hazard ratio; iCR, incomplete revascularization; POCO, patient oriented composite outcome.

**Figure 3 jcm-09-00232-f003:**
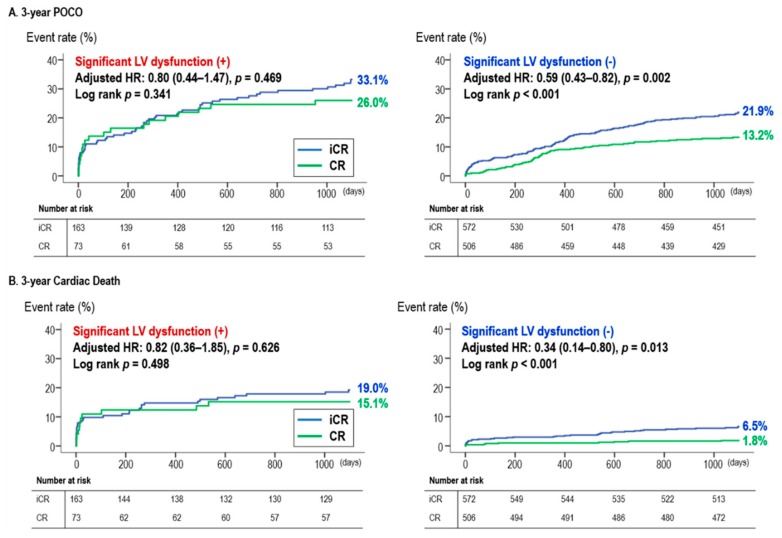
Survival curves during the three-year follow up period according to the presence of LV dysfunction: (A) three-year POCO and (B) three-year cardiac death. Abbreviations: CR, complete revascularization; HR, hazard ratio; iCR, incomplete revascularization; LVEF, left ventricular ejection fraction; POCO, patient oriented composite outcome.

**Figure 4 jcm-09-00232-f004:**
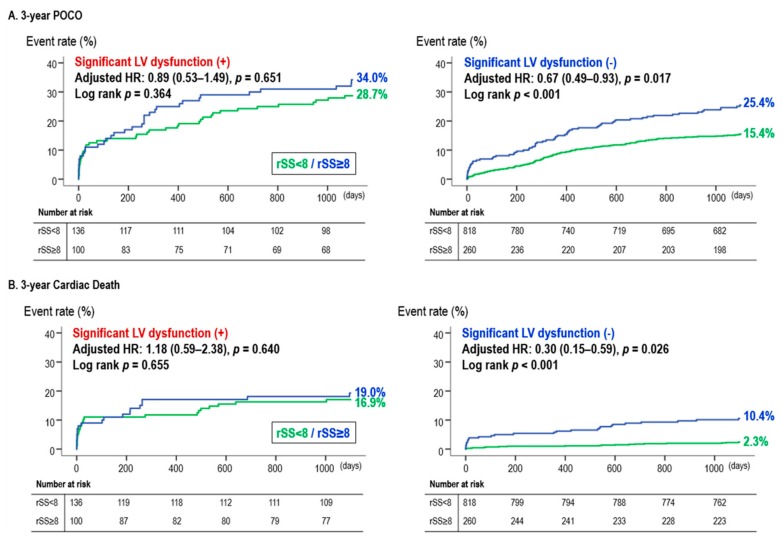
Survival curves during the three-year follow up period, according to the residual SYNTAX score: (**A**) three-year POCO and (**B**) three-year cardiac death. Abbreviations: HR, hazard ratio; LVEF, left ventricular ejection fraction; POCO, patient oriented composite outcome; rSS, residual SYNTAX score.

**Table 1 jcm-09-00232-t001:** Baseline characteristics according to complete revascularization.

	Total *(n* = 1314)	CR *(n* = 579)	iCR *(n* = 735)	*p*-Value
**Demographics**				
Age (years old)	63.3 ± 12.1	62.1 ± 11.8	64.2 ± 12.3	0.001
Male sex, *n* (%)	996 (75.8%)	452 (78.1%)	544 (74.0%)	0.101
Body mass index (kg/m^2^)	23.9 ± 3.0	24.0 ± 3.0	23.9 ± 3.0	0.540
Diabetes mellitus, *n* (%)	427 (32.5%)	195 (33.7%)	232 (31.6%)	0.451
Hypertension, *n* (%)	688 (52.4%)	281 (48.5%)	407 (55.4%)	0.016
Dyslipidemia, *n* (%)	607 (46.2%)	263 (45.4%)	344 (46.8%)	0.658
Current smoking, *n* (%)	633 (48.2%)	286 (49.4%)	347 (47.2%)	0.465
Previous stroke, *n* (%)	87 (6.6%)	36 (6.2%)	51 (6.9%)	0.682
Congestive heart failure, *n* (%)	20 (1.5%)	8 (1.4%)	12 (1.6%)	0.887
Chronic renal failure, *n* (%)	486 (38.6%)	194 (34.5%)	292 (42.0%)	0.007
Peripheral vascular disease, *n* (%)	14 (1.1%)	9 (1.6%)	5 (0.7%)	0.207
Prior MI, *n* (%)	93 (7.1%)	41 (7.1%)	52 (7.1%)	1.000
Prior PCI, *n* (%)	129 (9.8%)	63 (10.9%)	66 (9.0%)	0.291
Family history of CAD, *n* (%)	74 (5.6%)	30 (5.2%)	44 (6.0%)	0.611
**Angiographic findings**				
Angiographic disease extent				
2 vessel disease, *n* (%)	795 (60.5%)	407 (70.3%)	388 (52.8%)	<0.001
3 vessel disease, *n* (%)	519 (39.5%)	172 (29.7%)	347 (47.2%)	
Left main disease, *n* (%)	50 (3.8%)	24 (4.1%)	26 (3.5%)	0.670
Bifurcation lesion, *n* (%)	490 (37.3%)	246 (42.5%)	244 (33.2%)	0.001
Type B2/C lesion, *n* (%)	1117 (85.0%)	499 (86.2%)	618 (84.1%)	0.326
Calcified lesion, *n* (%)	109 (8.3%)	48 (8.3%)	61 (8.3%)	1.000
Tortuous lesion, *n* (%)	280 (21.3%)	136 (23.5%)	144 (19.6%)	0.100
Thrombus in lesion, *n* (%)	454 (34.6%)	206 (35.6%)	248 (33.7%)	0.524
Previously treated lesion, *n* (%)	106 (8.1%)	46 (7.9%)	60 (8.2%)	0.966
Culprit lesion, *n* (%)				0.787
LM, *n* (%)	36 (2.7%)	15 (2.6%)	21 (2.9%)	
LAD, *n* (%)	620 (47.2%)	281 (48.5%)	339 (46.1%)	
LCX, *n* (%)	151 (11.5%)	70 (12.1%)	81 (11.0%)	
RCA, *n* (%)	505 (38.4%)	212 (36.6%)	293 (39.9%)	
Stent diameter, mm	3.1 ± 0.4	3.0 ± 0.4	3.1 ± 0.4	0.088
Stent diameter <3 mm, *n* (%)	508 (38.7%)	224 (38.8%)	284 (38.6%)	1.000
Min. stent diameter, mm	3.0 ± 0.4	2.9 ± 0.4	3.0 ± 0.4	<0.001
Min. stent diameter <3 mm, *n* (%)	625 (47.6%)	296 (51.1%)	329 (44.8%)	0.025
Total stent length, mm	43.5 ± 25.9	49.9 ± 28.8	38.5 ± 22.1	<0.001
Total stent length ≥30 mm, *n* (%)	803 (61.2%)	398 (68.9%)	405 (55.1%)	<0.001
Total stent number	1.8 ± 1.0	2.1 ± 1.1	1.5 ± 0.8	<0.001
Staged PCI (among CR patients), *n* (%)	NA	97 (16.8%)	NA	NA
Second generation DES usage, *n* (%)	944 (71.8%)	417 (72.0%)	527 (71.7%)	0.947
Contrast volume, mL	272.8 ± 111.1	282.5 ± 107.6	263.6 ± 114.1	0.199
GP IIb/IIIa inhibitor usage, *n* (%)	143 (10.9%)	59 (10.2%)	84 (11.4%)	0.531
IVUS usage, *n* (%)	397 (30.2%)	201 (34.7%)	196 (26.7%)	0.002
Device success, *n* (%)	1290 (98.2%)	569 (98.3%)	721 (98.1%)	0.975
Lesion success, *n* (%)	1284 (97.7%)	564 (97.4%)	720 (98.0%)	0.634
Procedural success, *n* (%)	1281 (97.5%)	562 (97.1%)	719 (97.8%)	0.487
SYNTAX score at baseline	18.2 ± 8.8	15.6 ± 8.0	20.3 ± 8.9	<0.001
SYNTAX score after PCI (residual)	5.7 ± 6.4	1.7 ± 2.4	8.8 ± 6.8	<0.001
Delta SYNTAX score	12.5 ± 7.5	13.8 ± 7.8	11.5 ± 7.2	<0.001
**Laboratory data**				
LVEF (%)	50.7 ± 22.8	53.4 ± 31.7	48.7 ± 11.6	0.002
WBC (/ul)	10868 ± 3934	10715 ± 4075	10986 ± 3820	0.221
Hemoglobin (g/dL)	13.8 ± 2.1	13.9 ± 2.0	13.7 ± 2.2	0.071
Anemia (Hb <12 g/dL)	236 (18.2%)	93 (16.3%)	143 (19.6%)	0.148
Creatinine (mg/dL)	1.1 ± 0.7	73.5 ± 28.9	69.8 ± 30.1	0.027
Total Cholesterol (mg/dL)	181.5 ± 45.3	181.2 ± 45.1	181.8 ± 45.6	0.824
Triglyceride (mg/dL)	129.7 ± 87.8	133.1 ± 87.3	127.1 ± 88.1	0.260
HDL-cholesterol (mg/dL)	42.0 ± 12.1	41.9 ± 12.0	42.2 ± 12.2	0.685
LDL-cholesterol (mg/dL)	115.9 ± 38.0	116.3 ± 39.5	115.5 ± 36.9	0.722
**Discharge medication**				
Aspirin, *n* (%)	1306 (99.4%)	576 (99.5%)	730 (99.3%)	0.986
Clopidogrel, *n* (%)	1285 (97.8%)	569 (98.3%)	716 (97.4%)	0.389
DAPT, *n* (%)	1283 (97.6%)	568 (98.1%)	715 (97.3%)	0.429
Beta blocker, *n* (%)	1028 (78.2%)	460 (79.4%)	568 (77.3%)	0.380
ACE inhibitor or ARBs, *n* (%)	1049 (79.8%)	457 (78.9%)	592 (80.5%)	0.512
Statin, *n* (%)	1147 (87.3%)	512 (88.4%)	635 (86.4%)	0.310
Calcium channel blocker, *n* (%)	106 (8.1%)	51 (8.8%)	55 (7.5%)	0.439

Abbreviations: CR, complete revascularization; iCR, incomplete revascularization; MI, myocardial infarction; PCI, percutaneous coronary intervention; CAD, coronary artery disease; LM, left main; LAD, left anterior descending; LCX, left circumflex; RCA, right coronary artery; DES, drug-eluting stent; GP, glycoprotein; IVUS, intravascular ultrasound; LVEF, left ventricular ejection fraction; WBC, white blood cell; HDL, high-density lipoprotein; LDL, low-density lipoprotein; DAPT, dual antiplatelet agent; ACE, angiotensin-converting enzyme; ARB, angiotensin receptor blocker.

**Table 2 jcm-09-00232-t002:** Three-year clinical outcomes according to the LV function.

	Total	CR	ICR	Unadjusted		Multivariable-Adjusted		PSM		IPTW	
				HR (95% CI)	*p*-Value	HR (95% CI)	*p*-Value	HR (95% CI)	*p*-Value	HR (95% CI)	*p*-Value
**Reduced LV function**	*n* = 236	*n* = 73	*n* = 163								
POCO	73 (30.9%)	19 (26.0%)	54 (33.1%)	0.78 (0.46–1.31)	0.343	0.80 (0.44–1.47)	0.469	1.03 (0.50–2.14)	0.937	0.94 (0.66–1.36)	0.778
All cause death	51 (21.6%)	15 (20.5%)	36 (22.1%)	0.93 (0.51–1.70)	0.811	1.05 (0.51–2.15)	0.899	1.01 (0.42–2.44)	0.975	1.11 (0.72–1.70)	0.650
Cardiac death	42 (17.8%)	11 (15.1%)	31 (19.0%)	0.79 (0.40–1.57)	0.498	0.82 (0.36–1.85)	0.626	0.54 (0.18–1.65)	0.280	0.58 (0.34–1.01)	0.052
MI	15 (6.4%)	3 (4.1%)	12 (7.4%)	0.28 (0.04–2.22)	0.227	0.35 (0.03–4.17)	0.403	0.32 (0.01–70.87)	0.680	0.72 (0.28–1.88)	0.502
TVMI	12 (5.1%)	3 (4.1%)	9 (5.5%)	0.45 (0.05–3.85)	0.466	0.38 (0.01–24.31)	0.646	-	0.982	1.00 (0.30–3.39)	0.989
Stent thrombosis	7 (3.0%)	3 (4.1%)	4 (2.5%)	1.65 (0.37–7.39)	0.511	4.27 (0.54–33.80)	0.169	6.20 (0.18–215.37)	0.313	1.46 (0.28–7.62)	0.652
Any revascularization	23 (9.7%)	6 (8.2%)	17 (10.4%)	0.79 (0.31–2.00)	0.615	0.60 (0.21–1.76)	0.354	1.31 (0.31–5.64)	0.716	0.64 (0.33–1.25)	0.188
TLR	8 (3.4%)	4 (5.5%)	4 (2.5%)	2.27 (0.57–9.08)	0.246	3.07 (0.60–15.69)	0.178	-	0.850	3.77 (1.16–12.22)	0.027
TLF	46 (19.5%)	11 (15.1%)	35 (21.5%)	0.70 (0.36–1.38)	0.302	0.78 (0.35–1.72)	0.531	1.03 (0.39–2.71)	0.946	0.79 (0.50–1.27)	0.332
Any bleeding	3 (1.3%)	0 (0.0%)	3 (1.8%)	-	0.497	-	0.977	-	0.878	-	1.000
**Preserved LV function**	*n* = 1078	*n* = 506	*n* = 572								
POCO	192 (17.8%)	67 (13.2%)	125 (21.9%)	0.58 (0.43–0.78)	<0.001	0.59 (0.43–0.82)	0.002	0.60 (0.43–0.83)	0.002	0.66 (0.54–0.82)	<0.001
All cause death	82 (7.6%)	24 (4.7%)	58 (10.1%)	0.46 (0.29–0.74)	0.001	0.52 (0.30–0.89)	0.017	0.46 (0.26–0.83)	0.010	0.52 (0.51–0.99)	0.049
Cardiac death	46 (4.3%)	9 (1.8%)	37 (6.5%)	0.27 (0.13–0.56)	<0.001	0.34 (0.14–0.80)	0.013	0.22 (0.07–0.64)	0.006	0.50 (0.29–0.84)	0.001
MI	32 (3.0%)	10 (2.0%)	22 (3.8%)	0.39 (0.16–0.91)	0.030	0.40 (0.16–0.98)	0.045	0.33 (0.11–1.00)	0.051	0.67 (0.38–1.17)	0.154
TVMI	16 (1.5%)	6 (1.2%)	10 (1.7%)	0.42 (0.11–1.57)	0.196	0.50 (0.12–2.09)	0.341	0.62 (0.15–2.67)	0.523	1.13 (0.51–2.51)	0.758
Stent thrombosis	12 (1.1%)	5 (1.0%)	7 (1.2%)	0.80 (0.25–2.52)	0.701	1.26 (0.26–6.08)	0.770	1.35 (0.29–6.24)	0.701	1.56 (0.61–3.99)	0.348
Any revascularization	108 (10.0%)	43 (8.5%)	65 (11.4%)	0.73 (0.49–1.07)	0.102	0.69 (0.45–1.03)	0.071	0.65 (0.41–1.01)	0.056	0.65 (0.49–0.86)	0.003
TLR	36 (3.3%)	20 (4.0%)	16 (2.8%)	1.40 (0.72–2.70)	0.318	1.68 (0.79–3.55)	0.176	1.32 (0.63–2.76)	0.467	1.76 (1.05–2.94)	0.031
TLF	80 (7.4%)	28 (5.5%)	52 (9.1%)	0.60 (0.38–0.95)	0.029	0.75 (0.45–1.28)	0.296	0.64 (0.36–1.13)	0.124	0.93 (0.65–1.33)	0.698
Any bleeding	27 (2.5%)	10 (2.0%)	17 (3.0%)	0.65 (0.30–1.43)	0.286	0.86 (0.35–2.09)	0.735	0.62 (0.22–1.75)	0.362	0.69 (0.36–1.33)	0.265

Abbreviations: CR, complete revascularization; iCR, incomplete revascularization; HR, hazard ratio; CI, confidence interval; PSM, propensity score matched analysis; IPTW, inverse probability weighting analysis; LVEF, left ventricular ejection fraction; POCO, patient oriented composite outcome; MI, myocardial infarction; TVMI, target vessel myocardial infarction; TLR, target lesion revascularization; TLF, target lesion failure.

**Table 3 jcm-09-00232-t003:** Three-year POCO and cardiac death according to the residual SYNTAX score.

	Residual SYNTAX Score <8	Residual SYNTAX Score ≥8	*p*-Value	Multivariable Adjusted	
				Hazard Ratio (95% CI)	*p*-Value
**Three-year POCO**					
Total population	165/954 (17.3%)	100/360 (27.8%)	<0.001	0.71 (0.54–0.94)	0.017
Preserved LVEF	126/818 (15.4%)	66/260 (25.4%)	<0.001	0.67 (0.49–0.93)	0.017
Reduced LVEF	39/136 (28.7%)	34/100 (34.0%)	0.382	0.89 (0.53–1.49)	0.651
**Three-year Cardiac Death**					
Total population	42/954 (4.4%)	46/360 (12.8%)	<0.001	0.57 (0.35–0.94)	0.026
Preserved LVEF	19/818 (2.3%)	27/260 (10.4%)	<0.001	0.30 (0.15–0.59)	0.001
Reduced LVEF	23/136 (16.9%)	19/100 (19.0%)	0.679	1.18 (0.59–2.38)	0.640

Abbreviations: CI, confidence interval; POCO, patient oriented composite outcome; LVEF, left ventricular ejection fraction.

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
