# Peer review of "Complete Revascularization of Multivessel Coronary Artery Disease Does Not Improve Clinical Outcome in ST-Segment Elevation Myocardial Infarction Patients with Reduced Left Ventricular Ejection Fraction"

_jcm, 2020, doi:10.3390/jcm9010232_

Round 1

Reviewer 1 Report

Although an observational study with all its inherent limitations, it is well done, increasing our knowledge about complete revascularization after STEMI.

The moment of baseline echocardiography should be more exactly stated because it might influence the distribution of patients into the two groups of LV function. If echocardiography was performed after primary PCI, some improvement in LV function might have already happened. Therefore LV dysfunction group might include patients without potential for myocardial recovery after PCI, adding an additional bias to the final analysis.

Author Response

Response to Reviewer Comments

Reviewer:

Although an observational study with all its inherent limitations, it is well done, increasing our knowledge about complete revascularization after STEMI.

 [Response]

We sincerely thank the Reviewer #1 for his/her time and effort to review our manuscript. We are very happy that our manuscript was found interesting to Reviewer #1.

[Point 1]

The moment of baseline echocardiography should be more exactly stated because it might influence the distribution of patients into the two groups of LV function. If echocardiography was performed after primary PCI, some improvement in LV function might have already happened. Therefore LV dysfunction group might include patients without potential for myocardial recovery after PCI, adding an additional bias to the final analysis.

[Response]

Thank you for this sharp comment.

In this study, measurement of LVEF was performed when patients were clinically stable. LVEF assessment after primary PCI may not reliably predict chronic LV dysfunction, especially in STEMI patients.[23] Moreover, a recent study suggested that MI patients who have an improved LVEF during the follow-up may have better clinical outcomes than those with no improvement.[24] The absence of a follow-up LVEF assessment could act as a bias for our study.

Regarding this issue, we modified limitations sections of our manuscript.

Limitation section, line 297-301,

Finally, echocardiography was performed when the patient was clinically stabilized, before discharge. However, LVEF assessment immediately after PCI may not reliably predict chronic LV dysfunction, especially in STEMI patients.[23] Moreover, a recent study suggested that MI patients who have an improved LVEF during the follow-up may have better clinical outcomes than those with no improvement.[24] The absence of a follow-up LVEF assessment could act as a bias for our study.

23.       Stolfo, D.; Cinquetti, M.; Merlo, M.; Santangelo, S.; Barbati, G.; Alonge, M.; Vitrella, G.; Rakar, S.; Salvi, A.; Perkan, A., et al. St-elevation myocardial infarction with reduced left ventricular ejection fraction: Insights into persisting left ventricular dysfunction. A ppci-registry analysis. Int J Cardiol 2016, 215, 340-345

24.       Chew, D.S.; Heikki, H.; Schmidt, G.; Kavanagh, K.M.; Dommasch, M.; Bloch Thomsen, P.E.; Sinnecker, D.; Raatikainen, P.; Exner, D.V. Change in left ventricular ejection fraction following first myocardial infarction and outcome. JACC Clin Electrophysiol 2018, 4, 672-682.

Reviewer 2 Report

Comparison of complete and incomplete revascularization in STEMI patients remains a challenging issue in clinical practice. The manuscript titled: Complete Revascularization of Multivessel Coronary Artery Disease Does Not Improve Clinical Outcome in ST-segment Elevation Myocardial Infarction Patients with Reduced Left Ventricular Function showed an interesting view of the prognostic impact of coronary revascularization in patients with reduced LVEF. The authors had good idea to focus on patients with STEMI and LVEF<40%. I have several major concerns that need to be addressed by the authors:

     1.The study is an observational analysis and not a randomized trial. The choice of the stent, predilatation, post-stenting adjunctive balloon inflation, and the use of intravascular ultrasound or glycoprotein IIb/IIIa inhibitors were all left to the operators' discretion. Adjunctive therapy post-PCI was also determined by the treating physician. These leaves open the possibility of selection bias and treatment bias. The iCR group had more risk factors, such as old age, hypertension and chronic renal failure, and showed a higher coronary complexity. I believe that is very difficult to conclude that the differences for POCO and cardiac death in patients with STEMI is related only to CR and iCR. The manuscript is an excellent observational study, but the results are not the consequence of the decision to use arbitrarily CR or iCR.

How many patients beneficiated of AICD? Did AICD influence the mortality after CR or iCR? Echocardiography was performed after the patient was stabilized, which may clinically be assumed as the point of recovery of myocardial stunning. We don’t know the real LVEF before PCI. It is correct to assume the title “STEMI patients with reduced LVEF”? How many patients presented no-reflow phenomenon? How many patients presented malign reperfusion lesions? In the study were included patients on clopidogrel. How important can be the impact of ticagrelor or prasugrel on the morbidity and mortality? Please comment.

     5.During revascularization, more drug eluting stents with a longer total length and a smaller minimal stent diameter were used in the CR group. The both are associated with higher risk of stent thrombosis/restenosis. Did these parameters influence the final results after 3 years?

Author Response

Response to Reviewer Comments

Reviewer:

Comparison of complete and incomplete revascularization in STEMI patients remains a challenging issue in clinical practice. The manuscript titled: “Complete Revascularization of Multivessel Coronary Artery Disease Does Not Improve Clinical Outcome in ST-segment Elevation Myocardial Infarction Patients with Reduced Left Ventricular Function” showed an interesting view of the prognostic impact of coronary revascularization in patients with reduced LVEF. The authors had good idea to focus on patients with STEMI and LVEF<40%. I have several major concerns that need to be addressed by the authors:

[Response]

We sincerely thank Reviewer #2 for using your valuable time and experience to review our manuscripts.

[Point 1]

The iCR group had more risk factors, such as old age, hypertension and chronic renal failure, and showed a higher coronary complexity. I believe that is very difficult to conclude that the differences for POCO and cardiac death in patients with STEMI is related only to CR and iCR. The manuscript is an excellent observational study, but the results are not the consequence of the decision to use arbitrarily CR or iCR.

[Response]

Thank you for this comment.

We very much agree with your opinion. The risk profile differed between the CR and iCR groups. Unless performed in a randomized clinical trial, this is an inherent issue in observational studies. Factors such as old age, hypertension and chronic renal failure were more common in the iCR group, and the coronary complexity was higher in the iCR group. To compensate for any potential confounding factors we performed a Cox proportional hazards regression model using a backward elimination algorithm, propensity-score matched analysis, and an inverse probability weighted propensity-score matched analysis. (Variables such as age, gender, body mass index, previous hypertension, previous diabetes mellitus, previous dyslipidemia, previous PCI, previous peripheral vascular disease, current smoking, previous chronic renal failure, presence of anemia, presence of LV dysfunction, baseline SYNTAX score and achievement of complete revascularization were added as covariates.) Although we used multiple statistical models to correct for possible biases and confounders, the possibility of unforeseen confounders affecting outcome cannot be completely ruled out.

Regarding this issue, we have mentioned in the limitations section of our manuscript.

Limitation section, line 292-293,

“Although we used multiple statistical models to correct for possible biases and confounders, the possibility of unforeseen confounders affecting outcome cannot be completely ruled out. Accordingly, our study results may not the sole consequence of CR or iCR.

We modified the methods of our manuscript.

Methods section, line 112-115,

“To estimate the independent factors on endpoints, a multivariable Cox proportional hazards regression model using a backward elimination algorithm and 0.05 as the significance level was performed. In addition, the Cox proportional hazard regression in a propensity-score matched cohort (standardized mean difference of variables was <10%) and inverse probability weighted (IPW) Cox proportional hazard regression were also performed.

[Point 2]

How many patients beneficiated of AICD? Did AICD influence the mortality after CR or iCR?

[Response]

Thank you for the comment.

The ‘Grand DES registry’ is a nation-wide registry of 55 centers in Korea. Unfortunately, AICD-related data was not collected in this registry. However, among the enrollment centers, we reviewed the medical record of all patients from one of the centers; Seoul National University Hospital which contributed 235 (17.9%) patients for this study. Only one patient from iCR group (120 patients) received and an ICD implantation, who was free from 3-year POCO.

Although, we could not evaluate the beneficial effect of ICD in LV dysfunction after STEMI, we think this is a very important issue that should be studied in consecutive researches.

[Point 3]

Echocardiography was performed after the patient was stabilized, which may clinically be assumed as the point of recovery of myocardial stunning. We don’t know the real LVEF before PCI. It is correct to assume the title “STEMI patients with reduced LVEF”?

[Response]

Thank you for this sharp comment.

In this study, measurement of LVEF was performed when patients were clinically stable. LVEF assessment after primary PCI may not reliably predict chronic LV dysfunction, especially in STEMI patients.[23] Moreover, a recent study suggested that MI patients who have an improved LVEF during the follow-up may have better clinical outcomes than those with no improvement.[24] The absence of a follow-up LVEF assessment could act as a bias for our study.

As per the reviewers comment, we modified our study title to as follows;

“Complete Revascularization of Multivessel Coronary Artery Disease Does Not Improve Clinical Outcome in ST-segment Elevation Myocardial Infarction Patients with Reduced Left Ventricular Ejection Fraction

Regarding this issue, we modified limitation section of our manuscript.

Limitation section, line 297-301,

Finally, echocardiography was performed when the patient was clinically stabilized, before discharge. However, LVEF assessment immediately after PCI may not reliably predict chronic LV dysfunction, especially in STEMI patients.[23] Moreover, a recent study suggested that MI patients who have an improved LVEF during the follow-up may have better clinical outcomes than those with no improvement.[24] The absence of a follow-up LVEF assessment could act as a bias for our study.

Stolfo, D.; Cinquetti, M.; Merlo, M.; Santangelo, S.; Barbati, G.; Alonge, M.; Vitrella, G.; Rakar, S.; Salvi, A.; Perkan, A., et al. St-elevation myocardial infarction with reduced left ventricular ejection fraction: Insights into persisting left ventricular dysfunction. A ppci-registry analysis. Int J Cardiol 2016, 215, 340-345 Chew, D.S.; Heikki, H.; Schmidt, G.; Kavanagh, K.M.; Dommasch, M.; Bloch Thomsen, P.E.; Sinnecker, D.; Raatikainen, P.; Exner, D.V. Change in left ventricular ejection fraction following first myocardial infarction and outcome. JACC Clin Electrophysiol 2018, 4, 672-682.

[Point 4]

How many patients presented no-reflow phenomenon? How many patients presented malign reperfusion lesions?

[Response]

Thank you for this comment.

We reviewed the total population to check for any PCI complications. The following table presents the TIMI flow grade after PCI. As can be seen, >95% of the study population achieved a TIMI flow grade 3 in both groups, while there was no difference of the TIMI flow after PCI in both groups. From this result, we can assume that the increased events of the iCR group were not determined by peri-procedural complications.

Table. TIMI flow grade of culprit artery after revascularization

Reduced LVEF group

Preserved LVEF group

Total (N=236)

CR

(N=73)

iCR

(N=163)

P Value

Total (N=1078)

CR

(N=506)

iCR

(N=572)

P Value

TFG

0.073

0.425

TFG 0, n (%)

4 (1.7)

0 (0.0)

4 (2.5)

1 (0.1)

0 (0.0)

1 (0.2)

TFG 1, n (%)

2 (0.8)

2 (2.7)

0 (0.0)

1 (0.1)

1 (0.2)

0 (0.0)

TFG 2, n (%)

4 (1.7)

2 (2.7)

2 (1.2)

4 (0.4)

1 (0.2)

3 (0.5)

TFG 3, n (%)

226 (95.8)

69 (94.5)

157 (96.3)

1072 (99.4)

504 (99.6)

568 (99.3)

Abbreviations: LVEF, left ventricular ejection fraction; CR, complete revascularization; iCR, incomplete revascularization; TFG, TIMI flow grade

This will be presented as a Reviewer-only response, however, if the reviewer insists, we will add this to our manuscript.

[Point 5]

In the study were included patients on clopidogrel. How important can be the impact of ticagrelor or prasugrel on the morbidity and mortality? Please comment.

[Response]

Thank you for this sharp comment.

Current guidelines preferentially recommend more potent new generation P2Y12 inhibitors such as ticagrelor and prasugrel in acute coronary syndrome (ACS). However, prasugrel and ticagrelor were both approved for use and covered by the nationwide insurance South Korea in 2013. Because our study population was enrolled until 2014, an absolute majority of the patients were prescribed with clopidogrel.

Regarding the reviewer’s comment on the impact of ticagrelor or prasugrel on the morbidity and mortality, we well acknowledge major RCTs that have shown survival benefit after using new generation P2Y12 inhibitors.[I,II] However, the clinical effect of these agents has not been so convincing in East Asian studies.[III-V]

We added the issue to the limitations section of our manuscript.

Limitations section, line 294-296,

Fourth, although current guidelines recommend potent new generation P2Y12 inhibitors such as ticagrelor and prasugrel in STEMI patients, a majority of our study population were prescribed with clopidogrel. This is due to the delayed approval of these agents in Korea, which was in 2013.

I.         Wiviott, S.D.; Braunwald, E.; McCabe, C.H.; Montalescot, G.; Ruzyllo, W.; Gottlieb, S.; Neumann, F.J.; Ardissino, D.; De Servi, S.; Murphy, S.A., et al. Prasugrel versus clopidogrel in patients with acute coronary syndromes. N Engl J Med 2007, 357, 2001-2015.

II.         Wallentin, L.; Becker, R.C.; Budaj, A.; Cannon, C.P.; Emanuelsson, H.; Held, C.; Horrow, J.; Husted, S.; James, S.; Katus, H., et al. Ticagrelor versus clopidogrel in patients with acute coronary syndromes. N Engl J Med 2009, 361, 1045-1057.

III.         Kang, J.; Park, K.W.; Palmerini, T.; Stone, G.W.; Lee, M.S.; Colombo, A.; Chieffo, A.; Feres, F.; Abizaid, A.; Bhatt, D.L., et al. Racial differences in ischaemia/bleeding risk trade-off during anti-platelet therapy: Individual patient level landmark meta-analysis from seven rcts. Thrombosis and haemostasis 2019, 119, 149-162.

IV.         Park, D.W.; Kwon, O.; Jang, J.S.; Yun, S.C.; Park, H.; Kang, D.Y.; Ahn, J.M.; Lee, P.H.; Lee, S.W.; Park, S.W., et al. Clinically significant bleeding with ticagrelor versus clopidogrel in korean patients with acute coronary syndromes intended for invasive management: A randomized clinical trial. Circulation 2019, 140, 1865-1877.

V.         Serebruany, V.L.; Tomek, A.; Pya, Y.; Bekbossynova, M.; Kim, M.H. Inferiority of ticagrelor in the philo trial: Play of chance in east asians or nightmare confirmation of plato-USA? Int J Cardiol 2016, 215, 372-376.

[Point 6]

During revascularization, more drug eluting stents with a longer total length and a smaller minimal stent diameter were used in the CR group. The both are associated with higher risk of stent thrombosis/restenosis. Did these parameters influence the final results after 3 years?

[Response]

Thank you for this comment.

We agree with the viewpoint that stents with a longer total length and a smaller minimal stent diameter associated with higher risk of stent thrombosis/restenosis. During revascularization, more drug eluting stents with a longer total length and a smaller minimal stent diameter were used in the CR group. These may associated more rates of stent thrombosis and in-stent restenosis. However, even though, our results showed that there was no significant differences between CR and iCR group. The incidence of events is shown in the following table.

Table. Incidence of stent thrombosis and TLR (adopted from Supplementary Table 1 of the main manuscript)

Total

N=1314

CR

N=579

iCR

N=735

P Value

Stent thrombosis, n (%)

19 (1.4)

8 (1.4)

11 (1.5)

1.000

TLR, n (%)

44 (3.3)

24 (4.1)

20 (2.7)

0.167

Abbreviations: CR, complete revascularization; iCR, incomplete revascularization; TLR, target lesion revascularization

By using a Cox proportional hazards regression model, we could find that both a longer total stent length and a smaller minimal stent diameter were neither independent predictors of stent thrombosis and TLR, as shown in the Table below. Moreover, to compensate for the angiographic findings that may have influence on 3-year clinical outcomes, the SYNTAX score was included in the Cox proportional hazards regression model of the original analysis.

Table. Analysis of the total stent length and minimal stent diameter as a predictor of stent-related outcomes.

Clinical Factors

Hazard Ratio

95% CI

P value

Stent thrombosis

Minimal stent diameter

1.01

0.92-1.12

0.775

Minimal stent diameter <3.0mm

1.01

0.41-2.48

0.986

Total stent length

1.00

0.98-1.02

0.756

Total stent length >30mm

0.88

0.36-2.20

0.790

TLR

Minimal stent diameter

0.95

0.90-1.01

0.082

Minimal stent diameter <3.0mm

1.64

0.90-2.99

0.107

Total stent length

1.01

0.99-1.02

0.138

Total stent length >30mm

1.02

0.56-1.88

0.943

Abbreviations: CI, confidence interval; TLR, target lesion revascularization